# Phosphatidylethanolamine Improves Postnatal Growth Retardation by Regulating Mucus Secretion of Intestinal Goblet Cells in Piglets

**DOI:** 10.3390/ani14081193

**Published:** 2024-04-16

**Authors:** Nan Wang, Chengming Wang, Ming Qi, Xingtong Lin, Andong Zha, Bie Tan, Yulong Yin, Jing Wang

**Affiliations:** 1College of Animal Science and Technology, Hunan Agricultural University, Changsha 410128, China; wangnan0317@stu.hunau.edu.cn (N.W.); wangcm1028@stu.hunau.edu.cn (C.W.); qmcharisma@sina.com (M.Q.); lin08232021@163.com (X.L.); zhaandong18@mails.ucas.ac.cn (A.Z.);; 2Yuelushan Laboratory, Changsha 410128, China; yinyulong@isa.ac.cn

**Keywords:** phosphatidylethanolamine, postnatal growth retardation, endoplasmic reticulum stress, intestinal barrier, piglets

## Abstract

**Simple Summary:**

Phosphatidylethanolamine (PE) is a multifunctional phospholipid, which is abundant in intestinal cells and mucus. It provides mechanical protection to intestinal epithelial cells and underlying tissues, and is essential for the growth and development of newborn individuals. In this study, PE was supplemented to healthy piglets and postnatal growth retardation (PGR) piglets during lactation and after weaning to explore whether PE can alleviate the growth retardation of PGR piglets and its internal mechanism. The results showed that PE could increase the number of goblet cells and MUC2 secretion by regulating the differentiation of goblet cells, ER stress and apoptosis of small intestine, which ultimately improved the intestinal morphology and mucus synthesis and secretion function of PGR piglets and increased their growth performance.

**Abstract:**

Phosphatidylethanolamine (PE), a multifunctional phospholipid, is necessary for neonate development. This study aimed to explore the impact of the regulation of exogenous PE on postnatal growth retardation (PGR) by improving intestinal barrier function. Thirty-two neonatal pigs were divided into four groups according to their body weight (BW 2.79 ± 0.50 kg or 1.88 ± 0.40 kg) at 7 days old, CON-NBW, PE-NBW, CON-PGR, and PE-PGR. PE was supplemented to NBW piglets and PGR piglets during lactation and post-weaning periods. Compared with the NBW piglets, the growth performance of PGR piglets was lower, while PE improved the poor growth performance. PGR piglets showed injured intestinal morphology, as evidenced by the reduced ratio of villus height to crypt depth (VH/CD) and goblet cell numbers in the jejunum and ileum. PE recovered the intestinal barrier injury by increasing VH/CD and goblet cell numbers. The decreased MUC2 mRNA and protein expressions were observed in the small intestine of PGR piglets, and PE remarkably increased the expression of MUC2. Mechanistically, PE increased the goblet cell differentiation promoting gene *spdef* mRNA levels and reduced the mRNA expressions involved in endoplasmic reticulum stress in the jejunal and ileal mucosa of PGR piglets. Overall, we found that PE alleviated growth retardation by regulating intestinal health and generalized its application in neonates.

## 1. Introduction

Early intestinal development is essential for the growth and health of newborns [1]. At the early stage of life, the immune system of neonates is immature. Certain factors, such as invasion of exogenous pathogens, insufficient supply of nutrients, oxidative stress, and inflammation, can cause intestinal structure damage and abnormal barrier function [2,3,4]. Both structural and functional damage can slow down the growth rate of young animals and even lead to death. Studies have shown that the neonatal period is a critical period for mammalian intestinal health [5,6]. Thus, it is important to investigate effective early nutritional intervention to improve intestinal development during the neonatal period.

Phosphatidylethanolamine (PE), a multifunctional phospholipid, is necessary for neonates’ growth, accounting for 15~25% of the total phospholipids in mammalian cells [7]. It is abundant in enterocytes and mucus and provides mechanical protection to intestinal epithelial cells and underlying tissues [8]. PE can be extracted from soybeans, which have attracted much attention as a functional food in recent years. Soybeans are also rich in another functional lipid, phosphatidylcholine, which has been proven to play an important role in promoting neonatal brain development and preventing diseases [9]. PE has numerous biological activities, such as supporting the proliferation and differentiation of enterocytes, regulating lipid metabolism, affecting intestinal microbes, alleviating endoplasmic reticulum (ER) stress, participating in autophagy, etc. [10]. One study reported that exogenous PE supplementation could alleviate intestinal cell injury and inflammation in fish [11]. In addition, PE metabolism was found to affect T follicular helper cells’ differentiation and humoral immunity [12]. Ethanolamine is the head group of PE and can be used as a precursor in the synthesis of PE [13]. It was found that exogenous ethanolamine increased the content of PE in mammalian cells, regulated autophagy, and prolonged the lifespan of Drosophila melanogaster [14]. Furthermore, the addition of ethanolamine in moderate doses to the diet has been shown to improve gastrointestinal development and growth performance in weaning piglets [15]. However, the functions of PE and ethanolamine in intestinal epithelial cells may be different. Whether PE can regulate the early development of individuals is still unknown. Meanwhile, there are no animal and/or clinical trial data regarding the effect of PE on growth retardation.

As far as their intestinal anatomy and physiology, pigs are similar to humans, making them an ideal model for studying human intestinal health [16]. Postnatal growth retardation (PGR) piglets are defined as pigs with a normal birth weight (>1.1 kg) whose body weight (BW) is lower than about 70% of the average weight of healthy piglets at the same age (day 7) during postnatal growth and development under the same feeding conditions and without obvious trauma [17]. In the present study, PGR piglets were used as a model to simulate children with growth retardation. We supplemented PE to healthy piglets and PGR piglets during lactation and after weaning to explore whether PE could alleviate the growth retardation of PGR piglets and its internal mechanism, such as alleviating intestinal morphologic damage and restoring intestinal goblet cell function, to improve the productivity of the pig industry. We also consider the clinical significance of PE in targeting children with growth retardation and hope to promote the application of functional lipid PE in regulating the intestinal health of newborn animals or infants.

## 2. Materials and Methods

### 2.1. Animals and Experimental Design

Duroc × Landrace × Yorkshire crossbred piglets with the same genetic origin and maternal health condition were used in the present study. At 7 days of age, sixteen PGR piglets (BW 1.88 ± 0.40 kg) and sixteen healthy piglets (BW 2.79 ± 0.50 kg) that were pair-matched by litter were selected for the experiments. Piglets were randomly divided into four groups according to their BW, each with eight replicates, and with each replicate being one pig (*n* = 8): (1) CON-NBW group: normal BW piglets received normal saline; (2) PE-NBW group: normal BW piglets received PE; (3) CON-PGR group: PGR piglets received normal saline; (4) PE-PGR group: PGR piglets received PE. We purchased powdered PE (CAS: 39382-08-6, Hebei Pengyu Biotechnology Co., Ltd., Handan, China) extracted from soybean and mixed it with normal saline to make a suspension for easy feeding. Piglets received PE through oral administration at 0.78 g/d during the suckling period (from day 7 to 28) and 2.11 g/d during the post-weaning period (from day 29 to 49) (Figure 1). Other piglets were orally administered the same amount of normal saline. PE and normal saline were taken once a day in the morning. All piglets weaned at day 28. On day 49 of age, piglets were euthanized for sampling. According to the methodology used in previous studies, we first calculated the daily amount of ethanolamine required for piglets based on their feed intake, and then we calculated the corresponding amount of PE supplementation [9,18]. All piglets were housed in pens with hard, plastic, slatted flooring and had 24 h ad libitum access to feed and water throughout the study. After weaning, the piglets were raised in individual pens, and the feed formula for piglets is shown in Appendix A.

The BW, abdominal circumference (ACF), and crown–rump length (CRL) were measured at days 7, 17, 26, 33, 40, and 49. The average daily gain (ADG) and body mass index (BMI) were calculated as follows:ADG = (final weight − initial weight)/time (g/d)
BMI = BW/CRL^2^ (kg/m^2^)

### 2.2. Tissue Collection and Processing

On day 49 of the neonatal period, 5 mL of blood was collected aseptically from a jugular vein after overnight fasting and centrifuged at 3000× *g* for 10 min at 4 °C to obtain serum samples. Then, all piglets were euthanized through electrical shock to collect intestine tissues [19]. The heart, lung, spleen, liver, and kidney were obtained and weighed. The relative weight of all organs was calculated as the organ weight divided by the BW (g/kg). The jejunum and ileum were dissected and rinsed thoroughly with an ice-cold 0.9% sterile saline solution. The middle segments (2 cm) of the jejunum and ileum were cut and fixed in 4% formaldehyde for morphological evaluation. Samples of the jejunal and ileal mucosa were scraped and immediately snap-frozen in liquid nitrogen. All samples were stored at −80 °C until further analysis [20].

### 2.3. Serum Biochemical Index Assay

Biochemical parameters, including triglyceride (TG), total cholesterol (TC), high-density lipoprotein cholesterol (HDL-C), and low-density lipoprotein cholesterol (LDL-C), were detected in the serum samples using a biochemical analyzer (ZY-450, Shanghai Kehua Bio-Engineering Co., Ltd., Shanghai, China).

### 2.4. PE Concentration Analysis

The total lipid was isolated from the liquid-nitrogen-pulverized jejunal and ileal mucosa samples with the 5% Triton X-100 aqueous solution. The PE content in the jejunal and ileal mucosa was analyzed using a PE assay kit (ab241005, Abcam, Cambridge, UK) according to the manufacturer’s instructions.

### 2.5. Intestinal Morphology Detection

For morphological measurements, 4% formaldehyde-fixed jejunum and ileum tissues were sectioned and stained with hematoxylin and eosin (H&E). After dehydration, embedding, sectioning, and staining, morphological changes were examined under a Zeiss upright fluorescence microscope (Axio Scope 5, Carl Zeiss Co., Ltd., Oberkochen, Germany) [21]. The villus height and crypt depth were measured with ZEN 3.7 software.

For the goblet cell numbers, goblet cell staining was performed using the Periodic Acid-Schiff (PAS) Staining Kit (Beyotime Biotech. Inc., Shanghai, China) following the manufacturer’s instructions.

The sections of all piglets were used for H&E and PAS staining and data analysis.

### 2.6. Immunofluorescence Staining

The sections of the jejunum and ileum were blocked for 30 min at 37 °C with 5% bovine serum albumin (BSA) and then incubated with MUC2 antibody (1:200; Hunan Aifang Biotechnology Co., Ltd., Changsha, China) overnight at 4 °C. After washing three times with phosphate buffer saline (PBS) (pH 7.4), the slices were incubated with Cy3 conjugated Goat Anti-Rabbit IgG (1:400; Wuhan Service Bio Technology; Wuhan, China) for 30 min at 37 °C in the dark. The nuclei were stained with 40, 6-diamidino-2-phenylindole (DAPI) for 5 min and washed with PBS three times for 5 min per wash, then treated with a self-fluorescence quenching agent for 5 min. The images were obtained under a Zeiss upright fluorescence microscope. The sections of all piglets were used for immunofluorescence staining and data analysis.

### 2.7. RNA Extraction and Real-Time PCR

Total RNA was isolated from the liquid-nitrogen-pulverized jejunal and ileal mucosa samples with the RNAiso Plus reagent (Takara Biotechnology Co., Ltd., Dalian, China) according to the manufacturer’s protocol. The concentration and quality of the RNA were determined using the NanoDrop One Spectrophotometer (Thermo Fisher Scientific, Waltham, MA, USA). Complementary DNA (cDNA) was synthesized using the Evo M-MLV RT Mix Kit with gDNA Clean for qPCR (AG11728, Accurate Biotechnology (Hunan) Co., Ltd., Changsha, China) following the manufacturer’s instructions. The RT-qPCR was then performed on a LightCycler R480II Real-Time PCR Instrument (Roche, Basel, Switzerland) using the SYBR Green Premix Pro Taq HS qPCR Kit (AG11701, Accurate Biotechnology (Hunan) Co., Ltd., Changsha, China) in accordance with manufacturer’s instructions. The fluorescence PCR program was set as follows: pre-denaturation, 95 °C for 30 s, 1 cycle; PCR amplification, 95 °C for 5 s, 60 °C for 30 s, 40 cycles; melting, 95 °C for 5 s, 60 °C for 1 min, 1 cycle; cooling, 50 °C for 30 s. Primers used in the PCR assay are listed in Table 1. The relative expression of selected genes normalized by β-actin was calculated using the 2^−ΔΔCt^ method [22]. The data were expressed as relative values to those for the CON-NBW group.

### 2.8. Western Blot Analysis

The total protein was extracted from the jejunal and ileal mucosa using radio immunoprecipitation assay (RIPA) lysis buffer and quantified using a bicinchoninic acid (BCA) protein assay kit (Beyotime Institute of Biotechnology, Shanghai, China). Proteins were separated using 8–10% sodium dodecyl sulfate-polyacrylamide gel electrophoresis (SDS-PAGE) (Yeasen Biotechnology Co., Ltd., Shanghai, China) and transfected to PVDF membranes. Membranes were blocked with a rapid blocking solution (Beyotime Institute of Biotechnology, Shanghai, China) for 30 min and incubated overnight with primary antibodies, including spdef (1:1000; Hunan Aifang Biotechnology Co., Ltd., Changsha, China), GRP78 (1:1000; Hunan Aifang Biotechnology Co., Ltd., Changsha, China), bcl-2 (1:1000; Hunan Aifang Biotechnology Co., Ltd., Changsha, China), and β-actin (1:1000; Wuhan Service Bio Technology; Wuhan, China) at 4 °C, followed by incubation with secondary antibody horseradish peroxidase-conjugated goat anti-rabbit or mouse IgG (1:5000; ZSGB Biological Technology, Beijing, China) for 2 h at room temperature. All target protein measurements were quantified by measuring the intensity of bands using Alpha Imager 2200 Software (Alpha Innotech Corporation, San Leandro, CA, USA) and normalized to β-actin.

### 2.9. TUNEL Assay

TUNEL staining was performed using the Alexa 488 TUNEL Cell Apoptosis Detection Kit (Hunan Aifang Biotechnology Co., Ltd., Changsha, China), and images were obtained with a Zeiss upright fluorescence microscope. The apoptotic rate (%) of jejunal and ileal cells was measured according to the ratio of green-positive fluorescent cells to blue-labeled total cells [23].

### 2.10. Statistics

Two-way analysis of variance (ANOVA) was used to analyze the main effects of piglet status (i.e., NBW or PGR) and treatment (i.e., orally administered PE or not) and their interaction using the general linear model procedure of SPSS 20.0 statistical software (SPSS Inc., Chicago, IL, USA). When a significant interaction between piglet status and a treatment was found, a simple main effect analysis was performed, and the differences between 2 treatments across 2 piglet statuses were evaluated using independent *t* test. When the data did not follow a normal distribution, we used the Kruskal–Wallis method of nonparametric testing for analysis. Results are presented as means ± standard error of mean (SEM). *p*-values < 0.05 were taken to indicate statistical significance.

## 3. Results

### 3.1. PE Oral Administration Improved the Growth Performance of PGR Piglets

The growth curve of piglets is shown in the Figure 2. From day 7 to day 49, the average BW of PGR piglets was remarkably lower than that of NBW piglets (*p* < 0.05), and PE supplementation gradually increased the BW of NBW and PGR piglets (*p* < 0.05) (Appendix A). The ADG, ACF, CRL, BMI, and relative weight of all organs are shown in Table 2. Compared with NBW piglets, PGR piglets’ ADG, ACF, CRL, and BMI were reduced (*p* < 0.05), and PE supplementation improved the growth performance of piglets (*p* < 0.05). Meanwhile, PGR piglets showed higher relative weights of the heart and lung compared to NBW piglets (*p* < 0.05), and PE treatment decreased the relative heart weight of piglets (*p* < 0.05). No interaction effect between piglet status and diet was found in these indicators (*p* > 0.05).

### 3.2. PE Oral Administration Affected the Serum Lipid and Intestinal PE Levels in Piglets

The serum lipid and intestinal PE contents in piglets are shown in Figure 3. PGR piglets exhibited higher serum TG levels but lower PE concentrations in the jejunal and ileal mucosa compared to the NBW piglets (*p* < 0.05). PE supplementation decreased serum TG levels and increased PE levels in the small intestine of piglets (*p* < 0.05). There was interaction between piglet status and diet on the serum TG levels, which means that piglets’ status influenced the PE treatments’ efficacy (*p* < 0.05). PE treatment decreased the serum TG contents in PGR piglets but not in NBW piglets (*p* < 0.05). There were no notable differences in TC and HDL-C among the four groups (*p* > 0.05).

### 3.3. PE Supplementation Alleviated the Intestinal Morphological Injury of PGR Piglets

The ratio of villus height to crypt depth (VH/CD) and the mRNA gene expressions of ZO-1 in both the jejunal and ileal mucosa of PGR piglets were remarkably lower than those of NBW piglets (*p* < 0.05). PE supplementation increased the VH/CD and *ZO-1* mRNA levels in the jejunal and ileal mucosa of piglets and alleviated intestinal morphological injury (*p* < 0.05) (Figure 4A–C).

### 3.4. PE Supplementation Alleviated Intestinal Inflammation of PGR Piglets

Compared with NBW piglets, PGR piglets showed lower *IL-4* and *IL-10* mRNA levels, while they had higher *IFN-γ* mRNA levels in the jejunal and ileal mucosa (*p* < 0.05). PE treatment increased *IL-4* and *IL-10* mRNA levels in the jejunal and ileal mucosa and decreased *IFN-γ* mRNA levels in the ileal mucosa of piglets (*p* < 0.05) (Figure 4D). There were significant interactions between piglet status and diet on VH/CD and *IFN-γ* mRNA levels in the ileal mucosa, which means that PE supplementation significantly increased the VH/CD in the jejunum and ileum and decreased *IFN-γ* mRNA levels in the ileal mucosa of PGR piglets but not NBW piglets (*p* < 0.05).

### 3.5. PE Supplementation Increased the Number of Small Intestinal Goblet Cells and MUC2 Secretion of PGR Piglets

The number of goblet cells and the relative abundance of MUC2 in the jejunal and ileal mucosa of PGR piglets were extremely reduced (*p* < 0.05) in comparison with NBW piglets. PE supplementation increased the goblet cell numbers and MUC2 abundance of the jejunal and ileal mucosa of piglets (*p* < 0.05). There was interaction between piglet status and diet on the number of goblet cells, which means that PE supplementation significantly increased the number of goblet cells in the jejunum and ileum of PGR piglets but not NBW piglets (Figure 5A–D).

### 3.6. PE Supplementation Promoted the Differentation of Small Intestinal Goblet Cells of PGR Piglets

To further explore the possible reasons for the increase in the number of goblet cells and the secretion of MUC2, the goblet cell differentiation transcription factor *spdef* was detected in the jejunal and ileal mucosa of piglets. PGR piglets had lower *spdef* gene and protein abundances in the small intestine compared to NBW piglets, while PE supplementation enhanced the spdef protein abundance in the ileal mucosa and *spdef* mRNA expression in the jejunum and ileum of piglets (*p* < 0.05) (Figure 5E–G).

### 3.7. PE Supplementation Alleviated the Endoplasmic Reticulum Stress and Apoptosis in the Small Intestine of PGR Piglets

Impaired goblet cell mucin secretion may be associated with impaired ER function or its own apoptosis [24]. PGR piglets showed lower ER-stress-related gene *AGR2* and *EDEM1* mRNA expressions, but they had higher *GRP78*, *CHOP,* and *ATF6* mRNA expressions in the jejunal and ileal mucosa and *PERK* mRNA expressions in the jejunal mucosa compared to NBW piglets (*p* < 0.05). PE administration increased the gene expressions of *AGR2* and *EDEM1*, but it decreased the gene expressions of *GRP78* and *ATF6* in the jejunal or ileal mucosa (*p* < 0.05). Meanwhile, PE treatment reduced the gene expressions of *CHOP* in the jejunal mucosa and the *PERK* mRNA levels in the ileal mucosa of piglets (*p* < 0.05). There was interaction between piglet status and diet on the gene expressions of *AGR2* in the ileal mucosa (*p* < 0.05) (Figure 6A,B). At the same time, the protein expressions of GRP78 in the jejunum of PGR piglets were higher than those in NBW piglets, while the GRP78 protein abundances in the small intestine were reduced by PE treatment of piglets (*p* < 0.05) (Figure 6D–F).

In addition, PGR piglets had lower *bcl-2* mRNA levels but showed higher *caspase 3* mRNA levels in the small intestine (*p* < 0.05). PGR piglets also had lower bcl-2 protein abundances in the ileal mucosa compared to NBW piglets (*p* < 0.05). PE supplementation increased the bcl-2 gene and protein expressions and decreased the *caspase 3* mRNA levels in the jejunal and ileal mucosa of piglets (*p* < 0.05) (Figure 6C–F). Moreover, the TUNEL analysis indicated that the apoptosis rates of the jejunal and ileal mucosa of PGR piglets were notably higher than those of NBW piglets, while PE treatment reduced the apoptosis rates in piglets (*p* < 0.05). There was significant interaction between piglet status and diet on the apoptosis rates in the small intestine, which means that PE supplementation significantly increased the apoptosis rates of PGR piglets but not NBW piglets (*p* < 0.05) (Figure 7).

## 4. Discussion

Growth retardation usually occurs in an individual’s infancy and is accompanied by many adverse effects, such as metabolic disorders, systemic inflammation, or intestinal dysfunction [25]. At present, the problem of growth retardation in children is still quite serious in some poor, developing countries. In research targeting children in developing countries, nutrients in food, such as choline, vitamins, and trace elements, have been recognized as effective measures for the prevention and treatment of growth retardation [26]. Effective nutritional interventions might lead to compensatory growth in growth-retarded individuals [27]. In our study, PGR piglets had lower BW and ADG than CON piglets, whereas PE supplementation significantly improved the growth performance of PGR piglets, which was consistent with previous studies [28]. In addition, we found that the serum TG levels were higher but the intestinal PE contents were lower in PGR piglets, and serum cholesterol levels showed no significant difference, whereas PE treatment deceased the serum TG levels and increased the intestinal PE concentrations of PGR piglets. This might be explained by the fact that PE supplementation activates fatty acid oxidation in the intestine of PGR piglets to metabolize intestinal lipids and prevent excessive TG secretion into the circulation [29]. Wei Fang et al. demonstrated that PE supplementation could reduce the concentration of TG in the intestinal cells of the large yellow croaker and alleviate lipid metabolism disorder [11]. PE is found mainly in organs, such as the liver and intestine, and in the cell membrane and organelle membrane of the organism, and it can affect a variety of cellular processes as well as the stability and function of many membrane proteins. Major defects in PE metabolism during mammalian development may lead to individual death [13]. It has been confirmed that the deletion of the PCYT2 gene, the key cytidine transferase for PE synthesis, can lead to impaired muscle development and growth retardation in young individuals [30]. These results might indicate that the PE contents and its function are related to the regulation of growth and development in neonatal pigs.

In general, abnormal small intestinal development is thought to be associated with impaired growth performance. Consistent with previous studies, we observed that PGR piglets had impaired intestinal morphology and lower VH/CD and mRNA expression of *ZO-1* in the jejunal and ileal mucosa, which may be related to the abnormal development of the small intestine in PGR piglets [31]. Tight junction structures are an important part of the intestinal physical barrier, and ZO-1 is one of the tight junction proteins, which plays an important role in the distribution and maintenance of the tight junction [32,33]. In this study, we found that PE supplementation increased the VH/CD and mRNA expression of *ZO-1* in the jejunal and ileal mucosa of PGR piglets, improved intestinal morphological and structural damage, and may alleviate intestinal physical barrier damage. In addition, damage to the intestinal physical barrier may lead to increased intestinal permeability and trigger intestinal inflammation [34].

In this study, mRNA expression of anti-inflammatory cytokines IL-4 and IL-10 was significantly lower, and mRNA expression of pro-inflammatory cytokine IFN-γ was significantly higher in the jejunal and ileal mucosa of PGR piglets, suggesting a higher level of intestinal inflammation in PGR piglets. IL-4 is an immunomodulatory cytokine that plays an important role in stimulating M2 macrophage differentiation and mediating anti-inflammatory responses [35]. IL-10 can alleviate tissue inflammation by inhibiting an excessive inflammatory response, up-regulating innate immunity, and accelerating tissue repair mechanisms [36]. IFN-γ, a Th1-type cytokine, is related to the increased death of IECs and the destruction of the intestinal epithelial barrier function in inflammatory bowel disease [37]. Previous studies have also found low expression of IL-4 and IL-10 and increased concentration of IFN-γ in the intestine of IUGR piglets [38,39,40]. In the current study, PE could alleviate the intestinal inflammation by significantly increasing the mRNA expression of IL-4 and IL-10 and decreasing the mRNA expression of IFN-γ in the jejunal and ileal mucosa of PGR piglets. To sum up, PE has the effect of improving intestinal morphological damage and the inflammatory response of PGR piglets, which may be one of the reasons for compensatory growth in PE-supplemented PGR piglets.

The intestinal mucus layer is composed of mucin, antibacterial peptide, and lysozyme secreted by goblet cells, which are essential for maintaining intestinal homeostasis and resisting intestinal bacterial infection. They are a vital part of the intestinal chemical barrier [41]. The mucus secreted by goblet cells can promote the repair of damaged intestinal epithelial tissue, inhibit the proliferation of pathogenic bacteria, and maintain intestinal homeostasis [42]. Intestinal injury or inflammation may destroy the homeostasis of the intestinal mucus layer, damage the mucus protection function, and cause intestinal diseases [43]. In the present study, we found that the number of goblet cells in the small intestine and the mRNA and protein expression of MUC2 in the jejunal and ileal mucosa of PGR piglets were extremely low, which was in line with the results of previous studies in mice and IUGR piglets [44,45]. Goblet cells and their secretions are an important part of the intestinal mucus layer, and the reduction of goblet cells will lead to mucus layer dysfunction and destroy the integrity of the intestinal barrier [46]. MUC2 is secreted by goblet cells, which constitute the structural backbone of the intestinal epithelial mucus layer, and the abnormal expression of MUC2 is closely associated with the occurrence of intestinal diseases [47]. PE supplementation increased the number of goblet cells and the synthesis and secretion of MUC2 in the small intestine of PGR piglets.

There are two possible mechanisms to explain the increasing numbers of goblet cells and the secretion of MUC2. One is that PE might regulate the differentiation of stem cells to goblet cells, and another one is that PE might decrease the apoptosis of goblet cells. The transcription factor spdef, a goblet cell marker gene involved in the terminal differentiation of goblet cells, is reported to promote the number of intestinal goblet cells and the expression of MUC2, thus restoring the integrity of the mucus barrier [48,49]. PE supplementation increased the mRNA and protein abundances of spdef in the small intestine of PGR piglets, suggesting that PE may promote the differentiation of intestinal goblet cells, restore the function of the mucosal barrier, and thus increase the growth performance of PGR piglets.

An imbalance in phospholipids in intestinal epithelial cells will trigger ER stress and the unfolded protein response (UPR), leading to cell death of goblet cells [50]. The dimerization of MUC2 protein in the ER and subsequent O-glycosylation in the Golgi apparatus are the beginnings of mucus synthesis; after further oligomerization, mature mucins are stored as granules until they are released from goblet cells [51]. ER stress in goblet cells leads to the misfolding of MUC2 accumulated in the ER, resulting in the reduction of mucin and the abnormality of the mucus barrier [52]. A previous study has shown that the addition of PE to the culture medium can down-regulate the expression of genes related to the UPR pathway in fish intestinal cells and alleviate ER stress [11]. Goblet cells are extremely sensitive to ER stress; therefore, PE regulation of ER stress may be an important way to alleviate intestinal injury in PGR piglets [53]. AGR2, which is mainly expressed in goblet cells, is a member of the protein disulfide isomerase family, which contributes to protein folding in the ER and plays an important role in the processing and synthesis of MUC2 [54]. EDEM1 is an ER resident protein whose expression is controlled by the UPR, which accelerates the degradation of misfolded proteins and maintains ER homeostasis [55]. The UPR is initiated by the heat shock protein family molecular chaperone GRP78, which activates the key ER stress pathways IREα/XBP1, PERK/elF2α, and ATF6 [56]. Therefore, in this experiment, we analyzed the mRNA expression of *AGR2*, *EDEM1,* and four key ER stress protein sensors *GRP78*, *CHOP*, *PERK,* and *ATF6*. The results showed that PE supplementation alleviated ER stress and UPR in the small intestine of PGR piglets, promoted the correct folding of MUC2, restored the function of goblet cells, and may improve growth performance. Meanwhile, the activation of PERK and ATF6 could regulate CHOP and decrease the expression of the anti-apoptotic gene bcl-2, thereby initiating apoptosis [57]. Therefore, we further detected apoptosis-related genes and found that PGR significantly increased the expression of pro-apoptotic gene *caspase 3* and down-regulated the expression of anti-apoptotic gene *bcl-2*. PE supplementation reduced the apoptosis rate of small intestinal cells in PGR piglets, which was also confirmed by TUNEL apoptosis fluorescence staining. Therefore, PE alleviation of ER stress and apoptosis may be another mechanism to regulate the intestinal barrier function, especially the goblet cell function. However, in our current study, we have not yet been able to accurately verify whether PE alleviates the apoptosis of intestinal goblet cells, and further research is needed to verify this.

Although PGR piglets have a lower BW, resulting in a higher PE intake per kilogram of BW than NBW piglets, the PE intake of each group of piglets is within a reasonable range and will not have a significant impact on the experimental results [18]. Yang et al. treated IPEC-1 cells with 100 and 200 μM of ethanolamine and indicated that two dosages of ethanolamine had similar effects on the cell proliferation [58]. In addition, this study showed interactive effects on indicators, such as serum TG content, intestinal VH/CD, intestinal goblet cell number, and apoptosis rate. This may be due to intestinal dysplasia and lipid metabolism disorders in PGR piglets, and this needs to be verified by further studies [59].

## 5. Conclusions

In summary, we investigated the regulation effect of PE oral administration during lactation and post-weaning on the growth retardation of piglets. Our study revealed that PE could increase the number of goblet cells and MUC2 secretion by regulating the differentiation of goblet cells, ER stress, and apoptosis of the small intestine, which ultimately improved the intestinal morphology and mucus synthesis and secretion function of PGR piglets and increased their growth performance. These findings provide a nutritional intervention strategy for the prevention and treatment of growth retardation in neonatal animals or infants.

## Figures and Tables

**Figure 1 animals-14-01193-f001:**
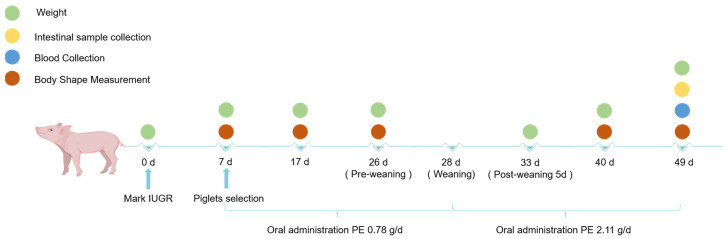
Animal experimental design for this study.

**Figure 2 animals-14-01193-f002:**
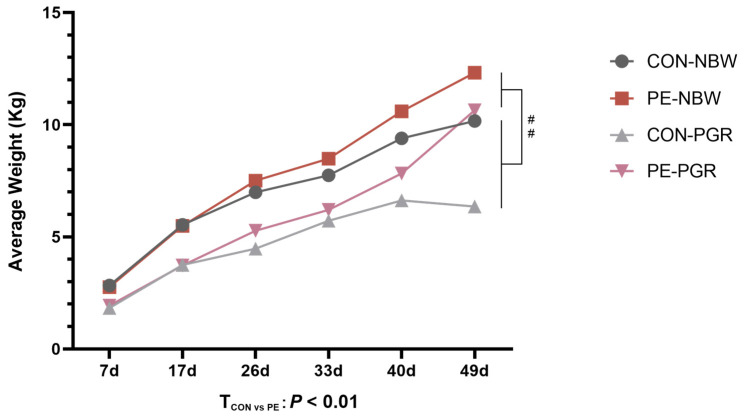
The growth curves of piglets in this study. Data are presented as mean ± SEM (*n* = 8). ^##^ indicates extremely significant difference between NBW and PGR groups (*p* < 0.01). T_CON vs. PE_: *p* < 0.01 indicates extremely significant difference between CON and PE groups.

**Figure 3 animals-14-01193-f003:**
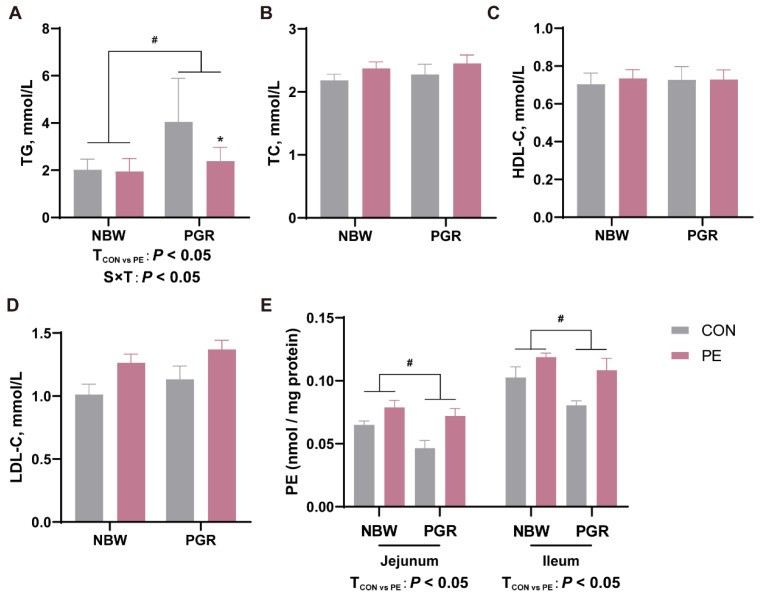
PE supplementation affected the levels of serum lipids and intestinal PE levels in PGR piglets. The concentrations of (**A**) triglyceride (TG), (**B**) total cholesterol (TC), (**C**) high-density lipoprotein cholesterol (HDL-C), and (**D**) low-density lipoprotein cholesterol (LDL-C) in the serum of piglets. (**E**) PE concentration in the jejunal and ileal mucosa. Data are presented as mean ± SEM (*n* = 8). ^#^ indicates significant difference between NBW and PGR groups (*p* < 0.05). T_CON vs. PE_: *p* < 0.05 indicates significant difference between CON and PE groups. S × T: *p* < 0.05 indicates an interaction between piglet status and diet. * indicates that the simple main effect analysis between 2 treatments across 2 piglet statuses is different. (*p* < 0.05).

**Figure 4 animals-14-01193-f004:**
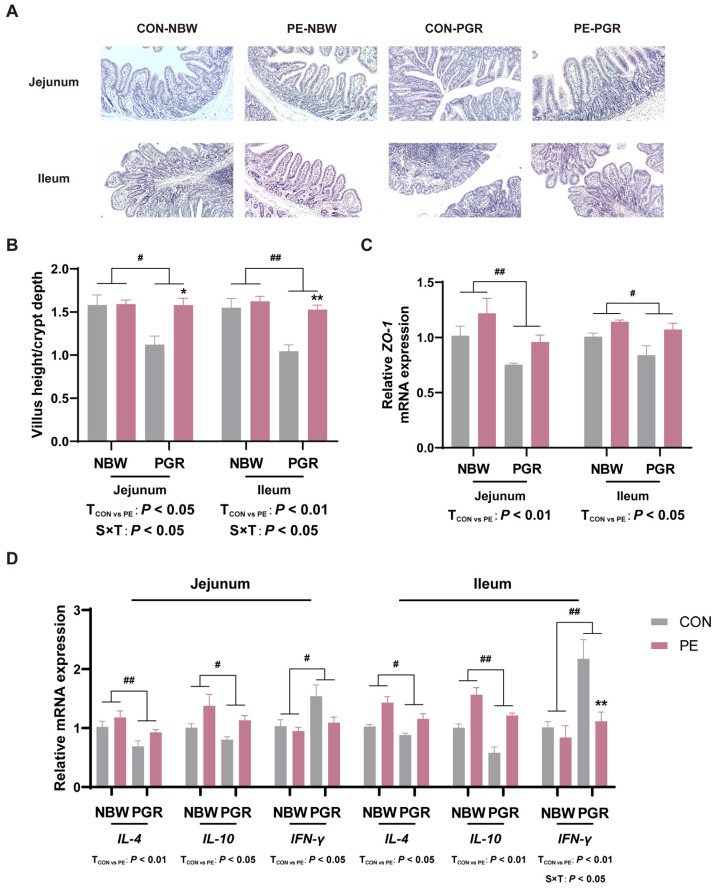
PE supplementation alleviated the intestinal injury and inflammation of PGR piglets. (**A**,**B**) Representative images of jejunum and ileum histology in piglets (magnification 100×) and the ratio of villus height to crypt depth (VH/CD). (**C**) The mRNA expression of *ZO-1* in the jejunal and ileal mucosa. (**D**) The mRNA expression of inflammatory factors *IL-4*, *IL-10,* and *IFN-γ* in the jejunal and ileal mucosa. Data are presented as mean ± SEM (*n* = 8). ^#^ indicates significant difference between NBW and PGR groups (*p* < 0.05). ^##^ indicates extremely significant difference (*p* < 0.01). T_CON vs. PE_: *p* < 0.05 indicates significant difference between CON and PE groups. T_CON vs. PE_: *p* < 0.01 indicates extremely significant difference. S × T: *p* < 0.05 indicates an interaction between piglet status and diet. * indicates that the simple main effect analysis between 2 treatments across 2 piglet statuses is different (*p* < 0.05). ** indicates extremely significant difference (*p* < 0.01).

**Figure 5 animals-14-01193-f005:**
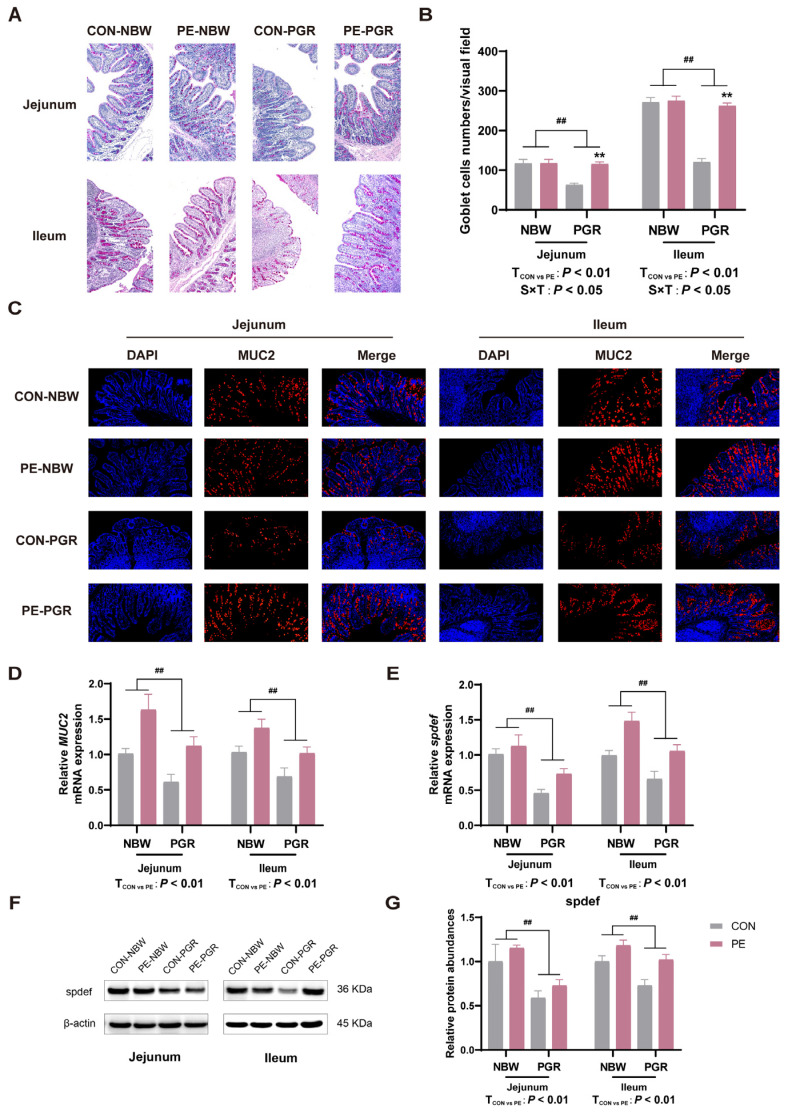
PE supplementation increases goblet cells’ differentiation and MUC2 synthesis in the small intestine of PGR piglets. (**A**,**B**) Representative images of Periodic Acid-Schiff (PAS) staining of the jejunum and ileum of piglets (magnification 100×) and goblet cell count. (**C**,**D**) Representative images of MUC2 immunofluorescence staining in the jejunum and ileum (magnification 200×) and its mRNA expression. (**E**–**G**) The mRNA and spdef in the jejunal and ileal mucosa. Data are presented as mean ± SEM (*n* = 8). ^##^ indicates extremely significant difference between NBW and PGR groups (*p* < 0.01). T_CON vs. PE_: *p* < 0.01 indicates extremely significant difference between CON and PE groups. S × T: *p* < 0.05 indicates an interaction between piglet status and diet. ** indicates that the simple main effect analysis between 2 treatments across 2 piglet statuses is extremely different (*p* < 0.01).

**Figure 6 animals-14-01193-f006:**
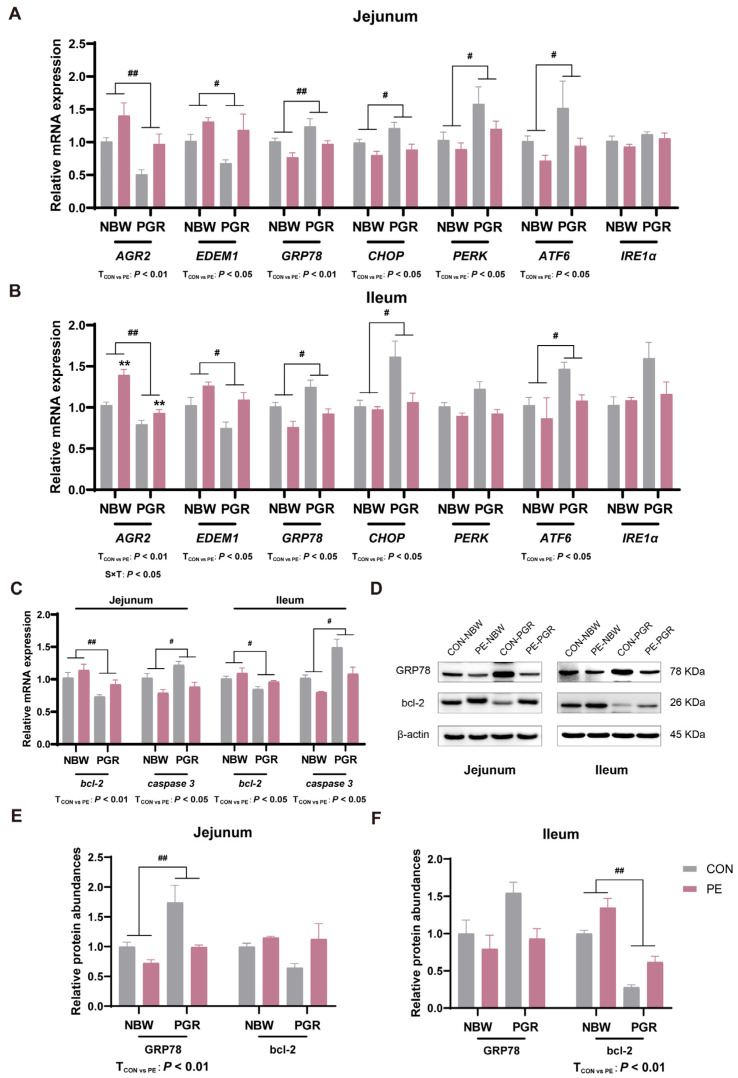
PE supplementation alleviated endoplasmic reticulum stress and apoptosis in the small intestine of PGR piglets. (**A**,**B**) The mRNA expression of ER-stress-related genes *AGR2*, *EDEM1*, *GRP78*, *CHOP*, *PERK*, *ATF6*, and *IRE1α* in the jejunal and ileal mucosa of piglets. (**C**) The mRNA expression of apoptosis-related genes *bcl-2* and *caspase3* in the jejunal and ileal mucosa. (**D**–**F**) The protein expression of GRP78 and bcl-2 in the jejunal and ileal mucosa. Data are presented as mean ± SEM (*n* = 8). ^#^ indicates significant difference between NBW and PGR groups (*p* < 0.05). ^##^ indicates extremely significant difference (*p* < 0.01). T_CON vs. PE_: *p* < 0.05 indicates significant difference between CON and PE groups. T_CON vs. PE_: *p* < 0.01 indicates extremely significant difference. S × T: *p* < 0.05 indicates an interaction between piglet status and diet. ** indicates that the simple main effect analysis between 2 treatments across 2 piglet statuses is extremely different (*p* < 0.01).

**Figure 7 animals-14-01193-f007:**
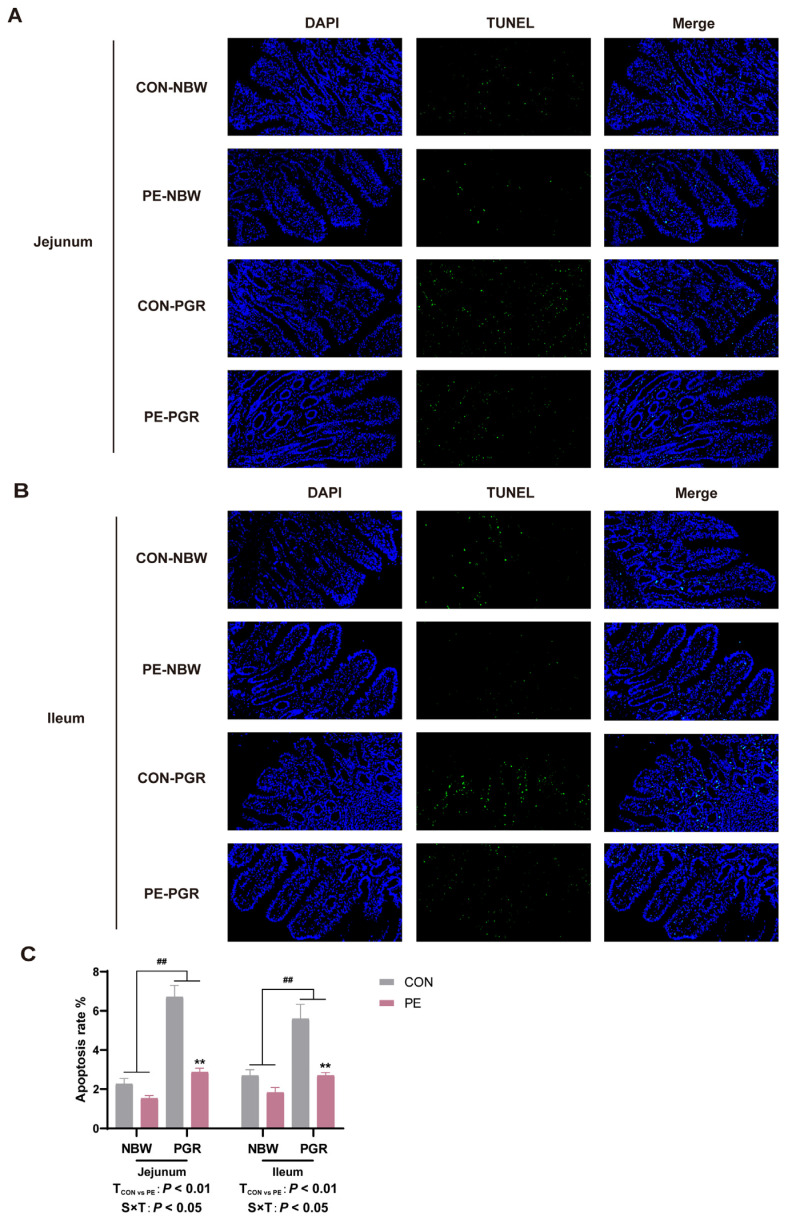
PE supplementation reduced the apoptosis rate of small intestinal cells of PGR piglets. (**A**–**C**) Representative images of TUNEL staining of the jejunum and ileum of piglets (magnification 200×) and the apoptosis rate. Data are presented as mean ± SEM (*n* = 8). ^##^ indicates extremely significant difference between NBW and PGR groups (*p* < 0.01). T_CON vs. PE_: *p* < 0.01 indicates extremely significant difference between CON and PE groups. S × T: *p* < 0.05 indicates an interaction between piglet status and diet. ** indicates that the simple main effect analysis between 2 treatments across 2 piglet statuses is extremely different (*p* < 0.01).

**Table 1 animals-14-01193-t001:** Primer sequences used in Real-Time PCR.

Gene	Gene Bank No.	Sequence (5′-3′)
*β-actin*	XM_0031242803	F: CTGCGGCATCCACGAAACT R: AGGGCCGTGATCTCCTTCTG
*ZO-1*	XM_021098896.1	F: CCTGAGTTTGATAGTGGCGTTGA
		R: AAATAGATTTCCTGCTCAATTCC
*IL-4*	NM_214340.1	F: CCCGAGTGTCAAGTGGCTTA R: TGATGATGCCGAAATAGCAG
*IL-10*	NM_214041.1	F: GGGCTATTTGTCCTGACTGC R: GGGCTCCCTAGTTTCTCTTCC
*IFN-γ*	NM_213948.1	F: TTCAGCTTTGCGTGACTTTG R: GGTCCACCATTAGGTACATCTG
*MUC2*	XM_021082584.1	F: CTGTGTGGGGCCTGACAA R: AGTGCTTGCAGTCGAACTCA
*spdef*	XM_005665895.3	F: GGCAGGGTTATGTGGGGAGTA R: GCTGTGTGAGGGGTGAGATAAT
*AGR2*	XM_005674278.2	F: AGCTCCTCCCTCTGTGTTAGG R: TGAGTATGTTCACCAGTGCCTT
*EDEM1*	XM_021069286.1	F: GGAAGGTCCCCAGCGTTTTA R: AAGACAAGCCACAGCACTCC
*GRP78*	XM_001927795.7	F: TATATAAGCGGAGCAGGCGAC R: TTCGCAAGCAAACCGATCAC
*CHOP*	XM_005674378.2	F: TCTGGCTTGGCTGACTGAGGAG R: CCGTTTCCTGGGTCTTCTTTGGTC
*PERK*	XM_003124925.4	F: ACTACAAGCGGGAAAGGAGC R: CACCAGTGCAAAAGGAGCAC
*ATF6*	XM_021089516.1	F: GCTCCTCCGTTCCTCCTTACCTC R: CTGACAACATGGGCTGCCTCTG
*IRE1α*	XM_005668695.3	F: CTGCACTCCCTCAACATCGT R: GTAGGTGGGGTTCTCCTTGC
*bcl-2*	NM_214285.1	F: AGGGCATTCAGTGACCTGAC R: CGATCCGACTCACCAATACC
*caspase 3*	NM_214131	F: CGTGCTTCTAAGCCATGGTG R: GTCCCACTGTCCGTCTCAAT

**Table 2 animals-14-01193-t002:** Growth performance and relative organ weights of piglets.

Item	Treatments		Piglet Status		SEM	*p*-Value		
	CON	PE	NBW	PGR		Treatment	Status	Treatment × Status
Average daily gain, g/d	158.16	226.34	223.75	160.75	8.00	<0.01	<0.01	0.127
Abdominal circumference, cm	42.71	49.22	48.65	43.28	0.72	<0.01	<0.01	0.192
Crown–rump length, cm	49.10	55.63	55.06	48.66	0.76	<0.01	<0.01	0.124
Body mass index, kg/m^2^	33.07	38.09	36.66	34.50	0.94	0.259	0.012	0.440
Relative weight, g/kg								
Heart	5.26	4.70	4.64	5.31	0.12	0.010	0.027	0.093
Liver	22.69	21.92	21.41	23.20	0.60	0.151	0.528	0.947
Spleen	1.89	1.67	1.68	1.89	0.08	0.185	0.171	0.527
Lung	17.83	13.73	15.57	16.02	0.89	0.799	0.026	0.421
Kidney	4.68	4.72	4.72	4.69	0.09	0.868	0.812	0.426

## Data Availability

The original contributions presented in this study are included in the article, and further inquiries can be directed to the corresponding author.

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
