# Peer review of "Phosphatidylethanolamine Improves Postnatal Growth Retardation by Regulating Mucus Secretion of Intestinal Goblet Cells in Piglets"

_animals, 2024, doi:10.3390/ani14081193_

Round 1

Reviewer 1 Report

Comments and Suggestions for Authors

Reviewer’s comments on the paper (Manuscript ID: foods-2834235) - Phosphatidylethanolamine improves postnatal growth retardation via regulating mucus secretion of intestinal goblet cells in piglets by Nan Wang et al.

Major comments:

This is a generally well-done and well-described study on the effects of oral administration phosphatidylethanolamine (PE) on growth and intestinal structure and function in postnatal pigs. The study appears to be one out of several similar papers on the same general theme. The study showed that feeding PE had a positive impact on growth performance and intestinal structure and function, e.g., mucus production and inflammatory response. This effect was of particular evident and of importance in post-natal growth retarded pigs.

Line 82-87; Please explain the basis for dosing of PE, ref. 17 referred to gives little help since they dosed phosphatidylcholine to babies. Moreover, if you calculate the given dose PE per body weight you get about 0,30 g/kg for the NBW piglets while about 0,50 g/kg for the smaller PGR piglets. Does this higher (about 40-50%) PE dose influence the results and account for the larger effects seen in the PGR group as compared to the NBW pigs?

Minor comments:

Line 76-80; To simplify for the reader use only one and the same acronym (not 2 different) for each exptl. group., eg, CON-NBW (CN) for normal BW (control) pigs receiving saline. This should also be applied in several of the figures (1B, 3A, 4A;F, 5D, 6A;D).   

Line 127; Specify the MUC2 antibody (the company – Hunan Aifang Biotechnology Co., Ltd is not searchable on the internet to this reviewer). Have any antibody specificity controls been done?

Line 157,159 and 160; Please specify antibody source and antibody specificity controls (see comment above)!

Line 198; Write the full parameter names in table text instead of acronyms to facilitate for the reader.

Line 199; Remove ADG = average daily gain!

Line 345,361 and 424; You should be careful of using the wording ‘intestinal physical barrier damage’ or ‘mucosal barrier function’ since you have not tested the intestinal barrier (permeation) properties in the study. 

Fig 4C and especially fig 6A;B; The images were difficult to evaluate!

Author Response

Response to Reviewer 1 Comments

Thank you so much for your letter and for the reviewers’ comments concerning our manuscript entitled “Phosphatidylethanolamine improves postnatal growth retardation via regulating mucus secretion of intestinal goblet cells in piglets” (foods-2834235). Revisions in the manuscript are highlighted in yellow. The corrections in the paper and the responses to the editor’ and reviewers’ comments are as follows.

Comments 1: Line 82-87; Please explain the basis for dosing of PE, ref. 17 referred to gives little help since they dosed phosphatidylcholine to babies. Moreover, if you calculate the given dose PE per body weight you get about 0,30 g/kg for the NBW piglets while about 0,50 g/kg for the smaller PGR piglets. Does this higher (about 40-50%) PE dose influence the results and account for the larger effects seen in the PGR group as compared to the NBW pigs?

Response 1: Thank you. According to the method in reference 9 (after revised), we calculated the corresponding amount of PE supplementation based on the amount of ethanolamine supplemented to piglets in Yang et al. study (reference 18). We first calculate the daily amount of ethanolamine required for piglets based on their feed intake. PE contains approximately 8% ethanolamine, and then calculate the daily dose of PE supplementation for piglets, which was also applied to another study of our group (DOI: 10.1093/lifemeta/load052/7564812, accepted). Line 92-94.

In addition, in our previous study (data not published, under review), we found that the plasma levels of glycerophospholipids, especially PE, were significantly lower in PGR piglets (around 60% of those in NBW pigs), so we supplemented 40% higher PE levels in PGR group to make up for the missing portion.

Comments 2: Line 76-80; To simplify for the reader use only one and the same acronym (not 2 different) for each exptl. group., eg, CON-NBW (CN) for normal BW (control) pigs receiving saline. This should also be applied in several of the figures (1B, 3A, 4A; F, 5D, 6A; D).

Response 2: Thank you for your good suggestion. We agree with you. For the convenience of readers, we have added captions below every image, CN: CON-NBW, PN: PE-NBW, CP: CON-PGR, PP: PE-PGR. Line 212-213, 233, 260-261, 289-290, 326-327, 335-336.

Comments 3: Line 127; Specify the MUC2 antibody (the company – Hunan Aifang Biotechnology Co., Ltd is not searchable on the internet to this reviewer). Have any antibody specificity controls been done?

Response 3: Thank you. The URL for this antibody is http://www.afantibody.cn/product_detail.aspx?id=195259&category_id=210

Antibodies used in this experiment were subjected to antibody specificity controls. The instructions for this antibody are shown in the attachment.

Comments 4: Line 157,159 and 160; Please specify antibody source and antibody specificity controls (see comment above)!

Response 4: Thank you. The URL for spdef antibody is http://www.afantibody.cn/product_detail.aspx?id=186358&category_id=210.

The instructions for spdef antibody are shown in the attachment.

The URL for GRP78 antibody is http://www.afantibody.cn/product_detail.aspx?id=187911&category_id=210.

The instructions for GRP78 antibody are shown in the attachment.

The URL for bcl-2 antibody is http://www.afantibody.cn/product_detail.aspx?id=196373&category_id=210.

The instructions for bcl-2 antibody are shown in the attachment.

Comments 5: Line 198; Write the full parameter names in table text instead of acronyms to facilitate for the reader.

Response 5: Thank you for your suggestion. We have revised. Table 2.

Comments 6: Line 199; Remove ADG = average daily gain!

Response 6: Thank you for your suggestion. We have revised. Table 2.

Comments 7: Line 345,361 and 424; You should be careful of using the wording ‘intestinal physical barrier damage’ or ‘mucosal barrier function’ since you have not tested the intestinal barrier (permeation) properties in the study.

Response 7: Thank you for your good suggestion. We agree with you. The FITC-dextran 4 is considered as the whole intestine permeability marker and is usually used in mice, while we did not conduct this intestinal permeability detection test in our piglets. Instead, we detected the tight junction protein ZO-1 expression and intestinal mucosal morphology to observe the barrier integrity. We have changed "intestinal physical barrier damage" to "intestinal morphological structure damage" and "mucosal barrier function" to "mucus synthesis and secretion function" in the whole manuscript. Line 374-375, 391-392, 462.

Comments 8: Fig 4C and especially fig 6A;B; The images were difficult to evaluate!

Response 8: Thank you. All the images of our same experiment were acquired under the same parameters, and the data were analyzed using ImageJ. The method of calculation is to obtain the area of DAPI staining (Area), and then obtain the integrated density of the positive area (IntDen), the average optical density (Mean) of each picture is calculated by the formula: Mean=Area/IntDen.

We tried our best to improve the manuscript and made some changes to the manuscript. We appreciate Editors/Reviewers’ warm work earnestly, and hope that the correction will meet with approval.

Yours sincerely,

Nan Wang

Reviewer 2 Report

Comments and Suggestions for Authors

The study conducted by Wang et al. evaluated of the alleviating effects of oral administration of phosphatidylethanolamine on postnatal growth retardation of piglets. The study is interesting and deals with the major problem of underdevelopment of the digestive system in newborns. The manuscript is well written. However substantial revision should be considered.

Comments:

L13: provide the initial wt. of animals.

L14: please provide full description of the experimental groups.

The abstract should be provided with the rearing condition and measures assessed.

All acronyms should be defined at first mention.

L60-64: contradiction. How growth-retarded and normal birth wt.?

L82: “Piglets received PE through oral administration at 0.78 g/d during the suckling period and 2.11 g/d during the post-weaning”.

What are the number of days for each period? Did PE was administered on a daily basis during these periods? was it administered as one dose or divided?

L92: the time should be clarified.

Sample size for each parameter should be provided.

The major concern of this study is number of animals and replication. Only one replicate for each group.

Provide the sample size of histological examination.

Table 2: as has been shown, the results are of the single effects. Please provide the results of the interaction.

Did the different periods of rearing affect the results? Did the authors measure the effect of PE and piglet status during each of these periods (suckling and after weaning)? as weaning is considered stress on the animals and affects the gut health so the period is considered a factor should be taken in consideration during analysis. If it isn’t a factor, why did the authors assess PE supplementation during these different periods?

L255, 256: spdef italic

Discussion: The authors should explain the PE effects on growth and correlate it with other measures.

Author Response

Response to Reviewer 2 Comments

Thank you so much for your letter and for the reviewers’ comments concerning our manuscript entitled “Phosphatidylethanolamine improves postnatal growth retardation via regulating mucus secretion of intestinal goblet cells in piglets” (foods-2834235). Revisions in the manuscript are highlighted in yellow. The corrections in the paper and the responses to the editor’ and reviewers’ comments are as follows.

Comments 1: L13: provide the initial wt. of animals.

Response 1: Thank you for your good suggestion. We have added the body weight of NBW and PGR piglets at the beginning of the experiment to the abstract. Line 14.

Comments 2: L14: please provide full description of the experimental groups.

The abstract should be provided with the rearing condition and measures assessed.

All acronyms should be defined at first mention.

Response 2: Thank you for your good suggestion. We agree with you, but considering the word limitation in the abstract (less than 200 words), we can only elaborate on the rearing condition and measures assessed of the experiment in detail in 2.1. Animals and Experimental Design.

Comments 3: L60-64: contradiction. How growth-retarded and normal birth wt.?

Response 3: Thank you. “Normal birth weight” refers to piglets with a birth weight greater than 1.1 kg. “Postnatal growth retardation (PGR) piglets” refer to piglets with a birth weight greater than 1.1 kg, but at 7 days of age, PGR piglets have a BW lower than 70% of other healthy piglets in the same litter, and have no obvious trauma. We have added the evaluation criteria for PGR piglets in the manuscript for better understanding. Line 61-65.

Comments 4: L82: “Piglets received PE through oral administration at 0.78 g/d during the suckling period and 2.11 g/d during the post-weaning”.

What are the number of days for each period? Did PE was administered on a daily basis during these periods? was it administered as one dose or divided?

Response 4: Thank you. Piglets received PE through oral administration at 0.78 g/d during the suckling period (from day 7 to 28) and 2.11 g/d during the post-weaning period (from day 29 to 49). Other piglets were oral administered with the same amount of normal saline. PE and normal saline were taken once a day in the morning. We have revised. Line 88-91.

Comments 5: L92: the time should be clarified.

Sample size for each parameter should be provided.

The major concern of this study is number of animals and replication. Only one replicate for each group.

Provide the sample size of histological examination.

Response 5: Thank you. “Time” refers to the entire trial period, i.e.:42 days.

There were 4 groups in this experiment, each with 8 replicates, each replicate being 1 pig, for a total of 32 pigs. The sample size for each parameter has been mentioned in each caption, i.e., n=8. We have revised. Line 81-82.

All 32 piglet sections were used for histological examination. Four fields of view per section were selected for data statistics and analysis.

Comments 6: Table 2: as has been shown, the results are of the single effects. Please provide the results of the interaction.

Response 6: Thank you. This study used a 2×2 two-factor experimental design with two factors: piglet status (NBW or PGR) and treatment (CON or PE). The last column in Table 2: Treatment × Status indicates the interaction P-value.

Comments 7: Did the different periods of rearing affect the results? Did the authors measure the effect of PE and piglet status during each of these periods (suckling and after weaning)? as weaning is considered stress on the animals and affects the gut health so the period is considered a factor should be taken in consideration during analysis. If it isn’t a factor, why did the authors assess PE supplementation during these different periods?

Response 7: Thank you. Based on the body weight of piglet in different periods (lactation and post-weaning), we supplement different amounts of PE to piglets in order to ensure that the amount of PE supplemented to piglets is sufficient and effective. In this study, we measured the body weight of piglets on d 7, 17, 26, 33, 40, and 49 of age to observe the observe the optimal weight-gain effect of PE on PGR piglets. On d 49 of age, the body weight of PGR piglets in PP group (PGR piglets with PE treatment) were significantly higher than those in PC group (PGR piglets without PE treatment), and the body weight of PGR piglets in PP group even chased after the piglets in CN group (NBW piglets). Thus, we chose d 49 of age to sacrifice the piglets and sampling.

All piglets in out study have suffered the weaning stress, and we did not observe the significant difference in body weight gain between NBW and PGR piglets (from d 26 to 33 of age: body weight gain in NBW group is 0.88 kg; body weight gain in PGR group is 1.09). So we did not consider weaning as a factor, and weaning stress had no different effects between NBW and PGR piglets and did not affect the PE treatment effect.

We supplemented Table S1: Body weight of piglets at different ages.

Comments 8: L255, 256: spdef italic

Response 8: Thank you. We have changed the gene indicating spdef to italics, while the spdef protein has not been changed. Line 278.

Comments 9: Discussion: The authors should explain the PE effects on growth and correlate it with other measures.

Response 9: Thank you for your good suggestion. We have added discussions on the correlation between growth performance and other measures in the Discussion section. Line 392-393, 419-420, 441-442.

We tried our best to improve the manuscript and made some changes to the manuscript. We appreciate Editors/Reviewers’ warm work earnestly, and hope that the correction will meet with approval.

Yours sincerely,

Nan Wang

Reviewer 3 Report

Comments and Suggestions for Authors

I have no major comments for this manuscript, which can proceed after making the changes indicated herebelow.

1.      The authors must describe clearly and in detail the objectives of the study.

2.      The authors must present convincing arguments regarding the clinical significance of their work.

3.      Table 1. Please present all the details of the PCRs, not only the primers.

4.      Figure 1, graph A: this should be added as a separate figure in Methods.

5.      Please insert the histological pictures as separate figures and please comment on the findings.

6.      I suggest to divide the discussion in two or three sub-sections for better flow of reading.

7.      The work is not justified well and leaves queries regarding the rationale of carrying out the study.

Author Response

Response to Reviewer 3 Comments

Thank you so much for your letter and for the reviewers’ comments concerning our manuscript entitled “Phosphatidylethanolamine improves postnatal growth retardation via regulating mucus secretion of intestinal goblet cells in piglets” (foods-2834235). Revisions in the manuscript are highlighted in yellow. The corrections in the paper and the responses to the editor’ and reviewers’ comments are as follows.

Comments 1: The authors must describe clearly and in detail the objectives of the study.

Response 1: Thank you for your good suggestion. We have described our research objectives in detail in Introduction section. “We supplemented PE to healthy piglets and PGR piglets during lactation and after weaning to explore whether PE could alleviate the growth retardation of PGR piglets and its internal mechanism, such as alleviating intestinal morphologic damage and restoring intestinal goblet cell function, to improve the productivity of pig industry. We also consider the clinical significance of PE in targeting children with growth retardation and hope to promote the application of functional lipid PE in regulating the intestinal health of newborn animals or infants.” Line 65-72.

Comments 2: The authors must present convincing arguments regarding the clinical significance of their work.

Response 2: Thank you. We have cited relevant references in Discussion section. “Nutrients rich in foods, such as choline, vitamins and trace elements, have been recognized as effective measures for the prevention and treatment of growth retardation. In the present study, PGR piglets can be used as a model to simulate children with growth retardation, we can also consider the clinical significance of PE in targeting children with growth retardation.” Line 342-346.

Comments 3: Table 1. Please present all the details of the PCRs, not only the primers.

Response 3: Thank you for your good suggestion. We have added PCR reaction conditions to the manuscript. “The fluorescence PCR program was set as follows: pre-denaturation, 95°C for 30s, 1 cycle; PCR amplification, 95°C for 5s, 60°C for 30s, 40 cycles; melting, 95°C for 5s, 60°C for 1min, 1 cycle; cooling, 50°C for 30s”. Line 158-160.

Comments 4: Figure 1, graph A: this should be added as a separate figure in Methods.

Response 5: Thank you for your good suggestion. We agree with your, but considering that separating figures would occupy a lot of space. We divided the histological pictures’ description into separate paragraphs in the Results section.

Comments 6:  I suggest to divide the discussion in two or three sub-sections for better flow of reading.

Response 6: Thank you for your good suggestion. We have split the Discussion section based on the modifications made in the Results section.

Comments 7: The work is not justified well and leaves queries regarding the rationale of carrying out the study.

Response 7: Thank you. As shown in the figure at the attachment, in our team's previous study (data unpublished, under review), we found that: the plasma levels of glycerophospholipids, especially PE, were significantly lower in PGR piglets as compared to NBW piglets. According to the previous result, we speculated that this could be a potential target for regulating the growth and development of PGR piglets. So, we attempted to explore whether oral administration of PE could improve the growth performance of PGR piglets in this study.

We tried our best to improve the manuscript and made some changes to the manuscript. We appreciate Editors/Reviewers’ warm work earnestly, and hope that the correction will meet with approval.

Yours sincerely,

Nan Wang

Reviewer 4 Report

Comments and Suggestions for Authors

The author studied phosphatidylethanolamine on intestinal histology and its effect on postnatal growth retardation in piglets. Results required to be reworked, especially in those traits consisting of interaction effect.  

Line 172: were data checked for normality? Was there a random factor in this experimental design?

Figure 1C: Since there was no interaction effect, ADG can be merged into Table 2. In addition, only the main effect should be presented without interaction effects. This rule should apply throughout the whole manuscript.

Line 223: PE supplementation on VH/CD only happened in PGR treatment. Here is a good example: when there are interaction effects, the discussion on the main effect can be misleading.

Author Response

Response to Reviewer 4 Comments

Thank you so much for your letter and for the reviewers’ comments concerning our manuscript entitled “Phosphatidylethanolamine improves postnatal growth retardation via regulating mucus secretion of intestinal goblet cells in piglets” (foods-2834235). Revisions in the manuscript are highlighted in yellow. The corrections in the paper and the responses to the editor’ and reviewers’ comments are as follows.

Comments 1: Line 172: were data checked for normality? Was there a random factor in this experimental design?

Response 1: Thank you. Most of our data fit a normal distribution (as shown in the attachment), and some, such as the relative weights of the lungs, did not. When the data do not follow a normal distribution, use the Kruskal Wallis method of nonparametric test for analysis. We have revised. Line 194-196.

Meanwhile, our study was conducted in the same pig farm, and each litter was screened 4 piglets to separate to each group (CN, PN, CP, PP). The genetic background, environmental conditions, feeding management methods among all piglets were the same, the independent variables in our study were PE treatment and piglet status. Thus, we ignore the influence of random factors.

Comments 2: Figure 1C: Since there was no interaction effect, ADG can be merged into Table 2. In addition, only the main effect should be presented without interaction effects. This rule should apply throughout the whole manuscript.

Response 2: Thank you for your suggestion. We have merged ADG into Table 2.

In addition, this study used a 2×2 two-factor experimental design with two factors: piglet status (NBW or PGR) and treatment (orally administered PE or not). While investigating the effect of PE on PGR piglets, it is important to investigate whether PE has the same effect on NBW piglets and to analyze the possible reasons for this. Therefore, we believe it is necessary to present the interaction effects.

Comments 3: Line 223: PE supplementation on VH/CD only happened in PGR treatment. Here is a good example: when there are interaction effects, the discussion on the main effect can be misleading.

Response 3: Thank you. Our study was 2 × 2 two factors experimental design. In the Results section, we describe the main effect, like piglet status and PE treatment, at first. If there were interaction effect, we describe the different effect of PE on NBW piglets or PGR piglets, and if there was no interaction effect, we declared “no interaction effect between piglet status and diet was found in these indicators” (normally in the end of each paragraph). In order to avoid the misleading of the main effect, we have revised the discussion, please review Line 453-456.

We tried our best to improve the manuscript and made some changes to the manuscript. We appreciate Editors/Reviewers’ warm work earnestly, and hope that the correction will meet with approval.

Yours sincerely,

Nan Wang

Round 2

Reviewer 1 Report

Comments and Suggestions for Authors

Comments after author revision:

The changes done in the manuscript according the editor/reviewer´s comments and suggestions have been addressed and are mainly satisfactory. The revised manuscript has gained in clarity and relevance.

(Major comment)

Thank you for answer to my major comments, however, it would be advantageous if you also could include the basis for the dosing in the manuscript and not only leave the reader with the references!

You should also discuss in the manuscript the possible impact of the different dosing to the two weight groups of pigs.     

Minor comments:

Line 81-85: Though now better explained in the running text and figure texts the authors still use double acronyms for the different experimental groups (e.g., CN group: normal BW piglets received normal saline (CON-NBW)! This does not improve the readability and does not appear to be necessary and this reviewer would prefer, e.g., CON-NBW for the group - controls with normal birth weight?

Author Response

Response to Reviewer 1 Comments

Thank you so much for your letter and for the reviewers’ comments concerning our manuscript entitled “Phosphatidylethanolamine improves postnatal growth retardation via regulating mucus secretion of intestinal goblet cells in piglets” (animals-2834235). Revisions in the manuscript are highlighted in yellow. The corrections in the paper and the responses to the editor’ and reviewers’ comments are as follows.

Point-by-point response to Comments and Suggestions for Authors:

Comments 1: It would be advantageous if you also could include the basis for the dosing in the manuscript and not only leave the reader with the references!

You should also discuss in the manuscript the possible impact of the different dosing to the two weight groups of pigs.

Response 1: Thank you for your good suggestion. We agree with you. We have added the basis for the dosing in the manuscript. Line 92-94.

In addition, we discussed the possible impact of PE dosage on the experimental results in the Discussion section of the manuscript. Although PGR piglets have a lower BW, resulting in a higher PE intake per kilo-gram of BW than NBW piglets, the PE intake of each group of piglets is within a reasonable range and will not have a significant impact on the experimental results. Yang et al. treated IPEC-1 cells with 100 and 200 μM ethanolamine and indicated that two dosage of ethanolamine had similar effects on the cell proliferation (reference 58). Line 446-450.

Comments 2: Line 81-85: Though now better explained in the running text and figure texts the authors still use double acronyms for the different experimental groups (e.g., CN group: normal BW piglets received normal saline (CON-NBW)! This does not improve the readability and does not appear to be necessary and this reviewer would prefer, e.g., CON-NBW for the group - controls with normal birth weight?

Response 2: Thank you for your good suggestion. We agree with you. We have revised the manuscript and figures. Line 81-85, 163. Figure 2, 4, 5, 6, 7.

We tried our best to improve the manuscript and made some changes to the manuscript. We appreciate Editors/Reviewers’ warm work earnestly, and hope that the correction will meet with approval.

Yours sincerely,

Nan Wang

Reviewer 2 Report

Comments and Suggestions for Authors

Thank you for the revision.

No further comments 

Author Response

Response to Reviewer 2 Comments

Thank you so much for your letter and for the reviewers’ comments concerning our manuscript entitled “Phosphatidylethanolamine improves postnatal growth retardation via regulating mucus secretion of intestinal goblet cells in piglets” (animals-2834235).

We would like to thank you for your professional review work, constructive comments,and valuable suggestions on our manuscript.

Yours sincerely,

Nan Wang